# Infant-Directed Speech to Preterm Infants during the First Year Postpartum: The Influence of Preterm Birth Weight and Maternal Parenting Stress

**DOI:** 10.3390/healthcare12030401

**Published:** 2024-02-04

**Authors:** Alessandra Provera, Erica Neri, Francesca Agostini

**Affiliations:** Department of Psychology, University of Bologna, 40126 Bologna, Italy; f.agostini@unibo.it

**Keywords:** infant-directed speech, prematurity, maternal parenting stress

## Abstract

Premature birth can increase the level of parenting stress (PS), especially in the case of parents of high-risk infants (extremely low birth weight (ELBW) and very low birth weight (VLBW)). Though published research has explored how maternal PS influences early dyadic interactions, limited research has focused on infant-directed speech (IDS), and no studies have investigated the link between prematurity severity based on birth weight and maternal IDS. This study, involving 100 mother–infant dyads, categorized into 30 ELBW premature infants, 30 VLBW premature infants, and 40 full-term (FT) ones, examined the impact of preterm birth weight and maternal parenting stress on IDS features during early interactions at 3 and 9 months postpartum. Maternal input was assessed using the CHILDES system, while parenting stress was evaluated using the Parenting Stress Index-Short Form. The results revealed that high-risk conditions (ELBW preterm birth and high parenting stress) at 3 months were associated with reduced affect-salient speech and increased questioning. IDS functional patterns, specifically the proportion of affect-salient speech and questions, were influenced by both birth weight groups and parenting stress levels at 3 months but not at 9 months. These findings highlight the need to assess, within the context of prematurity, both birth weight and parenting stress in clinical practice, offering insights for developing interventions supporting positive parent–infant interactions and facilitating infant development.

## 1. Introduction

Prematurity is defined as gestational age at birth < 37 weeks [1] and represents an important risk factor for the infant’s health and development [2,3], with an increased risk of both medical complications and the onset of deficits or delays which could affect developmental trajectories in different domains such as the cognitive, linguistic, and motor areas [4,5,6]. Problems in the acquisition of speech, communication, and language skills are more prevalent in the premature population compared to the full-term one [7]. These difficulties have been documented in the early stages of development, impacting, for example, the onset of gestural abilities [8], the acquisition of first words [9], and the development of early lexical and syntactic skills [9,10]. The risk of encountering delays or disorders in the developmental trajectories becomes more pronounced with lower gestational age [11] or when preterm infants are born with a very small birth weight, as in the case of very low birth weight (VLBW; BW < 1500 g) or extremely low birth weight (ELBW; BW < 1000 g) premature births [3]. Recently, a number of studies have highlighted the importance of investigating prematurity by specifically considering preterm birth weight, distinguishing between VLBW and ELBW preterm populations. In fact, although both VLBW and ELBW infants are classified as “high-risk” conditions due to their higher vulnerability to both mental and motor developmental disorders in the long term [12], several studies emphasize the importance of separately investigating the influence of preterm birth weight on parental outcomes, such as postpartum symptomatology, as well as early dyadic interactions. Indeed, the ELBW condition appears to be more frequently associated with higher levels of maternal depression, as well as the presence of atypical parent–infant interactive patterns [13,14,15].

As it constitutes a risk factor for the newborn’s survival and development, severe premature birth represents a stressful and even traumatic event also for parental well-being and mental health [13,14,16], impairing the processes of transition to parenthood [17] and increasing the risk of developing postpartum symptomatology (i.e., depressive, anxious, and PTSD symptoms), as well as parenting stress in both mothers and fathers during the first postpartum period [14,18,19,20,21].

The adjustment to the parenting role can be particularly challenging in the case of severe prematurity because parents must cope with concerns about both the infant’s health [22] and development [23], as well as their ability to effectively care for an infant often described as more easily irritable [24], more passive and less engaged in interactive exchanges [25,26,27], and less prone to express positive emotions [28]. Moreover, PT infants frequently experience prolonged hospitalization in a neonatal intensive care unit (NICU). The extended exposure to a noisy and overstimulating environment, such as that found in an NICU, coupled with the early separation from caregivers, may constitute an additional stress factor for both infants and their parents, also inhibiting early parent–infant communicative exchanges [29].

Consequently, parenting could represent a more stressful and demanding issue for PT infants’ mothers, increasing the risk of experiencing higher levels of psychological and parenting stress [30,31].

The presence of parental postpartum symptomatology and higher levels of parenting stress can adversely impact the quality of early dyadic interactions [27,32,33], including the quality of early dyadic exchanges between mothers and their infants, in terms of interactive behaviors [14,18,34,35,36] and verbal input directed towards the infant [15,37]. Consequently, these difficulties have the potential to result in more negative outcomes for infant development [34,38].

Literature that compared early dyadic behaviors in dyads of both PT and FT infants and their mothers highlighted significant associations between birth weight and the quality of mother–infant interactions during the first months postpartum. For instance, a study by Neri et al. [14] reported that mothers of more severe PT infants, specifically ELBW infants, were more likely to adopt intrusive behaviors during interactive exchanges, whereas VLBW mothers showed the opposite tendency, interacting with their infants in a more sensitive way. Furthermore, studies that investigated the effects of maternal parenting stress on the quality of early dyadic interactions highlighted that the more mothers were stressed, the more their interactive patterns tended to be intrusive, less attuned, and characterized by less positive interactions [35,36].

When talking to infants and young children, parents spontaneously use a linguistic register known as infant-directed speech [IDS], which differs from adult-directed speech [ADS] in terms of linguistic features, pragmatic functions, and acoustic patterns [39]. Verbal interactions directed towards infants are indeed characterized by a simpler lexicon and syntactic structure, frequent use of repetitions, more emphasized prosodic patterns, and specific functional and pragmatic features [40]. Infant-directed speech represents a fundamental aspect of caregiver–infant interactions, as its specific characteristics serve important functions by facilitating caregiver–infant interactions, capturing and sustaining the infant’s attention, and expressing both affective and informative contents [41,42,43,44]. Moreover, the features of IDS dynamically adapt over time, increasing in complexity to foster and facilitate the linguistic, cognitive, and socio-affective development of the infant [39,40,45].

Overall, IDS presents a reduced verbosity and frequency of speech compared to ADS [45]. Studies that analyzed IDS towards preverbal infants of different ages reported that adults tend to talk less with younger infants and to increase their speech verbosity when interacting with older infants [45,46,47]. Simplifications in the syntactic structure of IDS have been widely documented, with IDS presenting shorter and more syntactically simple utterances when compared with ADS [39,48]. Moreover, IDS addressed to preverbal infants is frequently characterized by single-word utterances and sentences without verbs [31,49]. Focusing on the lexical features, the literature on IDS has highlighted the use of a limited and redundant vocabulary characterized by a frequent use of repetitions. The lexicon used by caregivers while interacting with their infants is simpler compared to ADS and characterized by a recurrent use of diminutives, onomatopoeias, and nonsense words [41,45,46]. This aspect is more evident in younger infants and reflects the affect-salient function of IDS, which plays a fundamental role in facilitating affective exchanges in caregiver–infant interactions [39].

Several studies conducted on IDS have highlighted its dynamic nature, characterized by both changes and stability during the first postpartum years [40]. During the early stages of infant development, IDS is simpler in its linguistic component, and its pragmatic function is mainly affective. As the infant grows up, the verbal input becomes progressively more complex, and its pragmatic features become more informative rather than affect-salient to adapt to the child’s age and developmental stage [46,50]. Consequently, IDS features provide the infant with a linguistic and interactive environment that significantly influences the infant’s developmental trajectories across several domains, including linguistic, cognitive, and socio-affective development [39,45].

As previously mentioned, IDS represents a fundamental aspect in caregiver–infant exchanges, promoting closeness and the development of attachment bonding, as well as supporting the infant’s development in several domains. Concurrently, maternal emotional well-being and psychological stress have an influence on both formal and pragmatic aspects of maternal speech directed towards the infant [15,37,51]. Within this context, recent literature reported atypical IDS patterns related to both the presence of maternal postpartum symptomatology and parenting stress and the infant’s birth condition, specifically premature birth [15,37].

Although a broad range of articles underlined the influence of maternal symptomatology in the postpartum period on the quality of early interactions [13,14,19], the specific effects on the IDS pattern have not been adequately investigated. Specifically, even if the literature that focuses on mothers experiencing depressive symptoms has revealed variations in IDS patterns and characteristics when compared to non-depressed mothers’ speech [15,51], there is a notable lack of studies that have examined the connections between the quality of maternal speech directed to the infant and the presence of parenting stress. To the best of our knowledge, only a recent study by Spinelli and colleagues [37] investigated IDS towards 3-month-old PT and FT infants (corrected age for PT infants) while considering the levels of perceived parenting stress in mothers. The results highlighted that, although there was no main effect of prematurity on IDS characteristics, mothers who experienced higher levels of stress tended to use IDS that was more syntactically and lexically complex and thus more suitable for older infants. However, this study did not take into account the varying degrees of prematurity based on birth weight as a potential influencing factor, as well as subsequent stages of infant development.

Starting from these considerations, a longitudinal study was developed with the aim of investigating the impact of maternal parenting stress and the infant’s birth condition on the functional, syntactic, and lexical characteristics of IDS at 3 and 9 months postpartum. The term “postpartum” is employed to denote the chronological age in the case of FT infants. Conversely, for PT infants, each assessment considered the corrected age, determined by deducting the number of weeks the infant was born prematurely from his/her chronological age.

The first aim of the study was to investigate the presence of differences in the level of maternal parenting stress during the first 9 months postpartum. The second aim was to explore trajectories of stability or variation in IDS’s features from 3 to 9 months postpartum, also analyzing the presence of differences related to birth weight. Finally, the third and fourth aims were to explore if birth weight and/or maternal parenting stress at 3 months could have an influence on maternal speech features at 3 months postpartum and to longitudinally investigate if birth weight as well as levels of maternal parenting stress at both 3 and 9 months postpartum could influence IDS at 9 months.

We hypothesized that the presence of high-risk scenarios, such as prematurity associated with a lower birth weight and elevated levels of maternal parenting stress, might exert an influence on the features of maternal speech directed to the infant. Specifically, these circumstances may give rise to the development of different interactive patterns, in terms of differences in the syntactic, lexical, or functional attributes of maternal IDS.

## 2. Materials and Methods

### 2.1. Study Design

The present prospective observational cohort study is part of a larger research and follow-up project designed to explore the impact of prematurity on infant development, the occurrence of maternal and paternal symptomatology in the postpartum period, and the features of early interactive exchanges in the first 12 months postpartum of corrected age. This research project obtained approval from the Ethics Committee of the University of Bologna (Protocol Number 0001092/2023).

### 2.2. Participants

A total of 100 mother–infant dyads were recruited in this research. Among these, 40 mothers with their FT infants (gestational age > 36 weeks and birth weight > 2500 g) were enrolled during the third trimester of pregnancy at antenatal classes in Cesena. Additionally, 60 mothers with their PT infants (gestational age < 32 weeks and birth weight < 1500 g) who had been hospitalized in the Neonatal Intensive Care Unit (NICU) at Bufalini Hospital (Cesena, Italy) were recruited after hospitalization. The preterm dyads were further categorized into two subgroups based on infants’ birth weight: 30 preterm infants with birth weight between 1500 and 1000 g were classified into the very low birth weight (VLBW) group, and 30 preterm newborns with birth weight less than 1000 g constituted the extremely low birth weight (ELBW) group.

The exclusion criteria, which were the same for all the samples, were the existence of infant neurological disorders or complications, genetic syndromes or medical conditions, maternal pre-existing psychiatric conditions, and an absence of fluency in the Italian language.

During the study period, 137 preterm families were considered eligible for the study.

Among these, the families of 46 infants were excluded according to exclusion criteria, and 31 did not complete all study assessments, due to scheduling conflicts, leading to a final sample of 60 preterm infants.

### 2.3. Procedure

Data were collected at two different times of assessment, at 3 months (T1) and 9 months (T2) postpartum (corrected age was considered for PT infants), at the Laboratory of Developmental Psychodynamics (Department of Psychology, University of Bologna).

During each visit at 3 and 9 months, all participants were asked to provide their written consent and to complete questionnaires regarding sociodemographic characteristics and the levels of stress related to parental role. A psychologist also conducted assessments of the infant’s mental and psychomotor development by administering the Griffiths Mental Development Scales-Revised version (GMDS-R for 0–2 years; [52]). Following this, all mother–infant dyads were invited to participate in a 5 min free-interaction session. These assessments were conducted in a standardized setting which presented age-appropriate puppets and toys for the infants. Video recordings were made of each session, and subsequently, maternal speech directed towards the infant was fully transcribed in accordance with the Codes for the Human Analysis of Transcripts (CHAT) format [53]. This format is part of a broad project, the Child Language Data Exchange System (CHILDES) Project, and provides a standardized format for the generation of computerized transcription of face-to-face conversational interactive exchanges [54].

### 2.4. Measures

#### 2.4.1. Sociodemographic Data

The sociodemographic characteristics of the sample were investigated by administering an ad hoc sociodemographic questionnaire, which included information about maternal age and years of education, marital status, working condition, and parity, as well as perinatal data.

#### 2.4.2. Maternal Parenting Stress

The levels of maternal stress associated with the parental role were investigated by using the Italian-validated version [55] of the Parenting Stress Index-Short Form (PSI_SF) self-report questionnaire [56]. This questionnaire is widely used in international literature for the assessment of parenting stress, also in the context of preterm birth, and includes 36 items evaluated on a Likert scale of 1 to 5 points, with higher scores indicating the presence of higher levels of parenting stress. It also presents 3 subscales of 12 items investigating Parental Distress (PD), Parent-Child Dysfunctional Interaction (PCDI), and Difficult Child (DC) perception.

#### 2.4.3. Infant Mental and Psychomotor Development

As mentioned above, infant mental and psychomotor development at 3 and 9 months was assessed by individually administering the Griffiths Mental Development Scales-Revised version (GMDS-R for 0–2 years; [52]), specifically dedicated to the evaluation of infants and children between 0 and 2 years. Through a battery of developmental assessments, these scales estimate an infant’s global level of development represented by the mean score of five specific domains: locomotor skills, personal and social development, hearing and language proficiency, eye–hand coordination, and performance. The scores are standardized for an expected value of 100 with an SD of 12. For preterm groups, scoring always considered the corrected age at the time of assessment. The Griffiths scales are widely used for both clinical and research purposes to identify developmental delays and/or deficits and monitor infant’s acquisitions over time in high-risk contexts such as prematurity conditions. All infants were assessed by psychologists who were trained in the use of the GMDS and blind to infant birth weight.

#### 2.4.4. Infant-Directed Speech

Maternal IDS verbosity during the free-interaction session was investigated by considering the number of utterances addressed to the infant. The utterance was considered as the unit of analysis, and an “utterance” was defined as any continuous segment of speech separated from the subsequent one by a pause lasting more than 1 s [57].

To investigate the lexical and syntactic characteristics of maternal speech, the following measures were analyzed:Word tokens, which represent the total number of words produced.Word types, which represent the total number of different words produced.Mean Length of Utterance (MLU), a measure of syntactic complexity, calculated as the ratio of words to utterances.

Functional features of maternal IDS were analyzed through an ad hoc coding system that had been previously employed in maternal speech studies (e.g., [58,59]) and in a previous recent study on a preterm population [15]. In accordance with this scheme, each maternal utterance was assigned to one of the subsequent non-overlapping functional categories:Affect-salient speech, including utterances intended to sustain the conversation, such as greetings, words of encouragement, or singing.Information-salient speech, regarding utterances aimed at conveying information, either by providing or seeking it. It includes four subcategories: questions, labeling, descriptions, and directives.Attention-getter utterances, including utterances designed to capture the infant’s attention, such as calling the infant by their name.Other: incomplete or unintelligible utterances, as well as maternal speech not directed towards the infant.

We computed the proportion of maternal speech for each of these categories and subcategories by dividing the number of utterances falling into each category by the total number of maternal utterances produced during the interaction.

### 2.5. Data Analysis

Firstly, a series of analysis of variance (ANOVA) and Pearson’s chi-square tests were run to assess the homogeneity between the 3 groups according to mothers’ and infants’ sociodemographic features. More specifically, the groups were compared with respect to mothers’ age, years of education, marital status, parity, employment status, infant’s age (which was corrected for PT groups), and GMDS-R total score.

According to the first aim of the study, a repeated measures ANOVA was run in order to investigate the trajectories of the levels of maternal parenting stress from 3 to 9 months according to birth weight. For the second aim, a series of repeated measures ANOVAs were also run to explore trajectories of features of IDS across the first months postpartum.

Considering the third and fourth aims of our study, two series of multivariate regressions were run to identify possible predictors for the lexical, syntactic, and functional features of IDS at both 3 and 9 months. In the first series, we selected both birth weight and maternal PSI scores as predictors for maternal speech’s features at 3 months. The second series included both birth weight and maternal parenting stress at 3 and 9 months as predictors of maternal speech at 9 months. Given that birth weight presented three different conditions, we settled on ELBW as the reference group for the regression models and considered two different contrasts in our analysis: ELBW versus VLBW group (Birth weight 1) and ELBW versus FT group (Birth weight 2). This methodological step used to analyze the impact of a categorical variable with more than two levels is similar to criteria and analyses used in previously published articles (see, for example, [60,61]).

Data were analyzed using Jamovi software version 2.3.28 [62]. A *p* value < 0.05 was considered as statistically significant.

## 3. Results

### 3.1. Sociodemographic Characteristics

Descriptive features of mothers and infants included in the study are reported in Table 1. Statistically significant differences among groups were found in maternal years of education, parity condition, days of hospitalization after birth, infant’s developmental quotient at 3 months, and infant’s age at 9 months. Specifically, maternal years of education were higher in FT mothers compared to ELBW ones (Tukey post hoc: *p* = 0.002); infant’s gestational age at birth and birth weight were higher in the FT group compared to both the VLBW (Tukey post hoc: *p* = 0.010 and *p* = 0.001, respectively) and ELBW ones (Tukey post hoc: *p* = 0.010 and *p* = 0.001, respectively). Similarly, infant’s developmental quotient and infant’s age were higher in the FT group compared to the VLBW (Tukey post hoc: *p* = 0.010 and *p* = 0.001, respectively) and ELBW ones (Tukey post hoc: *p* = 0.001 and *p* = 0.001, respectively). Predictably, the hospitalization period prior to discharge was found to be longer in the ELBW group compared to the VLBW group and the term-born infant group (Tukey post hoc: *p* = 0.010 and *p* = 0.001, respectively). Moreover, multiparity was more frequent in the VLBW group compared to the FT and ELBW ones. Given that the distribution of these variables was not homogeneous, we ran a series of preliminary correlation analyses and ANOVAs to check their association with the dependent variables of the present study in order to control their possible influences. As we did not find any significant association, these variables were not included in subsequent statistical analyses.

### 3.2. Maternal Parenting Stress at 3 and 9 Months

Repeated measures ANOVA did not show any statistically significant effect within (F(2, 97) = 1.19; *p* = 0.307) or between (F(2, 97) = 0.2; *p* = 0.819) groups in the levels of maternal parenting stress during the first 9 months (Table 2).

### 3.3. Stability and Changes in IDS Features at 3 and 9 Months

Results of repeated measures ANOVAs (shown in Table 3) revealed an overall decrease in the number of questions (F(1, 97) = 12.40; *p* < 0.001) directed to the infant in all the three groups from 3 (M = 28.7; SE = 1.21) to 9 months (M = 21.4; SE = 0.89). A similar trend was found in the proportion of affective utterances, which appeared to be lower at 9 months (M = 43.0; SE = 1.47) compared to 3 months (M = 31.8; SE = 1.29) in all the groups (F(2, 97) = 43.90; *p* < 0.001). Nevertheless, when the effect of interaction between the birth group and the time of assessment was considered, we found significant differences that were specific only for ELBW dyads (F(2, 97) = 6.64; *p* = 0.002; Tukey post hoc, *p* = 0.775).

A statistically significant within effect was found for the number of both directives (3 months: M = 5.19; SE = 0.63; 9 months: M = 10.07; SE = 0.88; F(1, 97) = 21.92; *p* < 0.001) and attention-getter utterances (3 months: M = 3.78; SE = 0.52; 9 months: M = 10.21; SE = 0.75; F(1, 97) = 57.59; *p* < 0.001), which was higher at 9 months compared to 3 months in all the groups. A general increase was found also in the proportion of labels (3 months: M = 1.86; SE = 0.38; 9 months: M = 3.26; SE = 0.37; F(1, 97) = 5.60; *p* < 0.020), although the trend was statistically significant only in the FT group (Tukey post hoc: *p* = 0.005).

### 3.4. The Influence of Birth Weight and Maternal Parenting Stress on IDS Characteristics at 3 Months

Overall, all regression models aimed to test potential predictors of lexical and syntactic features of maternal speech at 3 months showed no statistically significant effect of either birth weight or parenting stress. Specifically, neither the number of word types (F(5, 94) = 2.44; *p* = 0.310), word tokens (F(5, 94) = 2.09; *p* = 0.062), nor MLU (F(5, 94) = 0.589; *p* = 0.708) was influenced by birth weight and maternal PSI scores at 3 months.

When models on functional features were investigated, regression analyses highlighted the presence of statistically significant equations only for the functional categories of affect-salient speech (F(5, 94) = 4.21; *p* = 0.002; Table 4, Figure 1) and questions (F(5, 94) = 4.19; *p* = 0.002; Table 4, Figure 2), which were both predicted by birth weight (specifically for the comparison between FT and ELBW groups), which was the main predictor according to β scores, and, to a lesser degree, by maternal parenting stress at 3 months, as well as their interactions. Specifically, maternal input directed towards ELBW infants was less affective and characterized by a higher use of questions compared to the FT group. Moreover, this trend appeared to be accentuated in mothers with higher levels of parenting stress, but only in the ELBW group (Figure 1 and Figure 2).

Conversely, the overall regression model did not result to be statistically significant in predicting the proportion of labels (F(5, 94) = 1.32; *p* = 0.263), descriptives (F(5, 94) = 1.34; *p* = 0.253), directives (F(5, 94) = 2.02; *p* = 0.083), or attention-getter utterances (F(5, 94) = 1.81; *p* = 0.119).

### 3.5. The Influence of Birth Weight and Maternal Parenting Stress on IDS Characteristics at 9 Months

Regarding the analysis of the predictors of lexical and syntactic features of maternal speech at 9 months, the number of word types (F(8, 91) = 0.97; *p* = 0.461), word tokens (F(8, 91) = 0.91; *p* = 0.509), and MLU (F(8, 91) = 1.15; *p* = 0.340) were influenced by birth weight and/or the levels of maternal parenting stress at 3 and 9 months.

Similarly, overall regression models on functional features assessing the proportion of directives (F(8, 91) = 0.62; *p* = 0.759), labels (F(8, 91) = 1.24; *p* = 0.284), descriptives (F(8, 91) = 1.51; *p* = 0.166), and attention-getter utterances (F(8, 91) = 0.62; *p* = 0.757) did not show statistically significant results.

In contrast, regression analyses focused on predicting affect-salient speech (F(8, 91) = 2.63; *p* = 0.012) and questions (F(8, 91) = 3.67; *p* < 0.001; Figure 3) produced statistically significant findings. Specifically, a lower proportion of affect-salient speech at 9 months appeared to be significantly predicted by higher levels of maternal parenting stress at 3 months; while the proportion of questions addressed to the infant was influenced by birth weight (being a mother of ELBW vs. FT infant), parenting stress at 3 months, and their interaction (higher levels of parenting stress were associated with a lower proportion of questions in FT and ELBW groups but not in the VLBW one; Figure 3). Among these, the predictor with higher β scores was birth weight, followed by the interaction between parenting stress at 3 months and the preterm birth weight, followed by PSI-SF scores at 3 months (Table 5).

## 4. Discussion

The general aim of this study was to investigate the potential influence of both preterm birth weight and levels of maternal parenting stress on infant-directed speech patterns during the first 9 months after childbirth.

### 4.1. Maternal Parenting Stress at 3 and 9 Months

Focusing on the first aim of our study, the results revealed the absence of statistically significant differences among groups in the maternal PSI scores at both 3 and 9 months. The literature on the impact of prematurity on parental stress levels has reported conflicting results to date. Indeed, while some studies indicated a significant impact of prematurity on maternal stress [30,32], others did not highlight differences between PT and FT populations [33,36,37,63]. Our finding aligned with this latter series of studies suggesting that, although premature birth could represent a stressful event for parents, PT infants’ mothers in our sample did not experience higher levels of parenting stress at both 3 and 9 months compared to FT ones. Moreover, previous studies have examined the potential relationship between maternal parenting stress and prematurity in the first trimester postpartum, finding no statistically significant differences associated with parenting stress levels. However, none of these studies employed a longitudinal design, and these studies did not explore the comparison between different levels of prematurity. Our study confirms and extends these findings also to the third trimester postpartum, further comparing ELBW and VLBW infants.

### 4.2. Stability and Changes in IDS Features at 3 and 9 Months

Overall, our findings revealed that the linguistic and structural aspects of maternal speech, specifically related to word types and tokens, as well as MLU, were not influenced by the birth condition at both 3 and 9 months. This outcome aligns with existing literature, which similarly observed no significant differences in lexical and syntactic features of maternal speech directed towards PT and FT infants [15,64,65]. Furthermore, our results confirmed findings from a previous study by Provera et al. [15], which, comparing the linguistic complexity of the maternal input directed to 3-month-old FT and PT infants and considering preterm birth weight, did not find any statistically significant difference between mothers of ELBW and VLBW infants, and extended this similarity also to later stages of development. The absence of significant differences in IDS complexity between 3 and 9 months may reflect a U-shaped trend in the trajectories of the linguistic characteristics of IDS during the first year. As suggested by several authors [45,66,67], there appears to be an initial increase in syntactic complexity during the first six months postpartum, followed by a subsequent decrease towards the end of that year and then by an increase during the second year postpartum. As mentioned above, this pattern reflects the dynamic nature of IDS which is deliberately adjusted by caregivers according to the emerging language abilities of the infant. In the early months, the heightened complexity of IDS may serve to stimulate the infant, while the observed decline in complexity towards the end of the first year may facilitate linguistic assimilation, thereby guiding and shaping linguistic development in a manner reflective of the infant’s growing capabilities [45].

The pragmatic features of IDS also appeared to be influenced by the infant’s age. At 3 months postpartum, maternal speech directed towards infants was predominantly characterized by affect-salient utterances, constituting more than half of the overall input directed to the infants; conversely, at 9 months, the proportion of affect-salient speech tended to decrease, and simultaneously, the number of labels, directives, and attention-getter utterances tended to increase. This IDS pattern is consistent with the existing literature which suggests that mother–infant communicative vocal exchanges during the first six months after birth are mainly affective and that a tendency to become more informative, descriptive, and contextualized [46,50,68] appears at later stages of development, when the infant starts to explore the surrounding environment and expresses a higher level of intersubjectivity [67]. The shift in the maternal communicative style is functional in supporting the infant’s emerging skills across the communicative, linguistic, and socio-cognitive domains [39]. In the present study, this trend seems to occur independently from the preterm status. Interestingly, the only exception regards IDS functional features directed to ELBW infants, specifically the proportion of affect-salient speech and labels, which appeared to be less influenced by the infant’s age. In fact, although maternal speech towards ELBW infants showed a little decrease in its affective features and an increase in the number of labels, this trend was not statistically significant, differently from what happens for FT infants. It could be hypothesized that mothers of PT infants with a lower birth weight may encounter challenges in adjusting their communicative style according to the infant’s age and developmental level, persisting in using an interactive style more suitable for younger infants. This result confirms the importance of exploring the effects of prematurity while also considering preterm birth weight, as highlighted by previous studies [13,14,15]. The way ELBW and VLBW mothers interact with their infants may be related to distinct representations of their infants and their capabilities. For example, preterm infants’ mothers may hold a stereotyped perception of their infant as more vulnerable, less mature, and less competent irrespective of the infant’s actual abilities or developmental level reached [69,70,71]. This representation, which could be more persistent in mothers of more vulnerable PT infants, like ELBW ones, may lead to the adoption of less suitable and functional interactive modalities for supporting the infants in their development.

### 4.3. The Influence of Birth Weight and Maternal Parenting Stress on IDS at 3 Months

In an exploration of the impact of parenting stress and birth weight on maternal input at 3 months, no significant effects were observed in relation to both syntactic and lexical features of maternal IDS. The lack of significant differences between groups in both lexical and syntactic complexity of IDS is consistent with previous results presented in a study by Spinelli et al. [37] which did not find differences in the linguistic complexity of maternal input directed towards PT and FT infants. Interestingly, in contrast to our findings, this study reported a significant influence of maternal parenting stress on these linguistic aspects of IDS (specifically in terms of MLU and the types/tokens ratio) at 3 months, reporting higher complexity of IDS associated with higher levels of maternal stress and thus suggesting the presence of less adequate interactive modalities. Again, this study compared FT and PT dyads, without taking into account preterm birth weight. Nevertheless, the differences in these findings underscore the necessity for further investigations keeping in mind the differentiation of preterm samples according to the levels of prematurity.

On the contrary, both the independent variables appeared to be significant predictors of IDS functional features at 3 months. Mothers of both ELBW and VLBW infants exhibited a tendency to talk with their infants in a more demanding way, using less affect-salient speech and using more questions compared to the FT group’s mothers. As mentioned above, maternal speech towards younger infants is expected to be more affective compared to IDS addressed to older ones. Various theoretical perspectives, including developmental psychology, psychopathology, infant research, and affective neurosciences, underscore the significant role of maternal affective speech in this early phase because it plays a crucial part in fostering positive interactions with the infant and facilitating the establishment of secure attachment bonding [13,72,73]. The tendency that emerged in our PT samples of addressing less affect-salient speech to younger infants became more pronounced with higher levels of parenting stress only for ELBW infants, suggesting that mothers in high-risk scenarios could experience more difficulties in establishing sensitive and attuned interactions with their infants. Literature on prematurity has frequently described PT infants as more passive and less engaged in dyadic interactions [74,75]. This characteristic could lead ELBW mothers to elicit the attention and engagement of their infants by addressing them with more questions during interactive exchanges. However, this interactive modality could also suggest a certain degree of intrusiveness in mother–infant interactions that is exacerbated in cases of mothers experiencing higher levels of parenting stress. Our hypotheses are consistent with a previous study by Muller-Nix et al. [75], which reported a higher frequency of intrusive behaviors and lower levels of sensitivity associated with more stressed mothers of preterm infants compared to FT and non-stressed ones. Moreover, our findings are also in line with previous studies which highlighted the presence of more controlling and intrusive behaviors in interactive exchanges with PT infants with a lower birth weight [13].

### 4.4. The Influence of Birth Weight and Maternal Parenting Stress on IDS at 9 Months

Consistent with our findings at 3 months, our findings indicated that neither birth weight nor parenting stress significantly influenced the lexical and syntactic characteristics of IDS at 9 months. This result could suggest a certain similarity in the structural and lexical complexity of maternal speech, which does not differ between FT and PT groups, aligning with previous findings from the literature [65]. Moreover, maternal parenting stress levels at both 3 and 9 months did not emerge as predictors of speech complexity at 9 months, therefore reinforcing the trend observed at 3 months also for later developmental stages. Concerning pragmatic features, our findings revealed that the proportion of affect-salient speech directed to the infants was significantly influenced by levels of maternal parenting stress at 3 months, but not at 9 months. Specifically, in all three groups, mothers experiencing higher stress at 9 months tended to engage with their infant using a less affective input.

Regarding the proportion of questions addressed to the infant, the results revealed that PT mothers compared to FT ones exhibited a less interrogative style while talking with their infants. Interestingly, the presence of parenting stress at 3 months appeared to influence the interactive style of ELBW and VLBW mothers differently. Similar to what happened with FT mothers, higher PSI scores at 3 months were associated with a lower number of questions directed to ELBW infants; conversely, VLBW mothers displayed the opposite pattern.

### 4.5. Limitations and Future Directions

While there is still a scarcity of literature examining IDS in the context of premature birth, and the available studies share comparable sample sizes with ours, one of the main limitations in our research is the restricted number of participants, which could limit the generalizability of our findings. Consequently, future studies should replicate this research with an increased number of participants involved. Our study focused on prematurely born infants with a gestational age < 32 weeks, categorizing them into two subgroups based on birth weight to evaluate potential significant differences between the ELBW and VLBW groups. While our primary emphasis was on birth weight, it is important to note that the exclusion of preterm infants with a gestational age > 32 weeks and a birth weight small for gestational age (SGA) represents a limitation in our study. This limitation restricts the generalizability of our results to this specific subgroup of preterm infants. Furthermore, it is essential to acknowledge that our study did not take into account other crucial variables affecting infants, such as prenatal steroids and various postnatal morbidities (e.g., intraventricular hemorrhage, bronchopulmonary dysplasia, and sepsis). These factors could significantly impact infant outcomes and contribute to increased parental stress during the postpartum period. A further limitation is related to the exclusion of fathers from this research design. Although the follow-up program provided by the Laboratory of Developmental Psychodynamics of the University of Bologna is addressed to both parents, the low participation rate of fathers precluded their inclusion in the present study. Subsequent studies should consider levels of parenting stress in both mothers and fathers, recognizing their potential role as significant risk or protective factors for child development and well-being [38]. Examining the influence of premature birth on the interactive behaviors of both parents can provide insights for tailoring interventions and support programs for families facing similar stressful circumstances. Lastly, in this study, we only considered the influence of parenting stress as a possible predictor for the characteristics of IDS patterns. However, given that existing literature indicates that parents of severely premature infants are at a heightened risk of experiencing various postpartum symptoms (e.g., depression, anxiety, PTSD symptoms), subsequent studies should explore the impact of different symptoms on maternal (and paternal) speech. Although promising, given the lack of other similar studies in the literature, our results should be replicated in further studies taking into account all these limitations.

## 5. Conclusions

Overall, the findings from this study support a potential connection between maternal psychological well-being and mental health in the first months postpartum and the quality of the interactive behaviors across the first year. Specifically, a significant relationship between maternal distress and linguistic interactive style has been explored and suggested.

The first period following childbirth represents a sensitive moment for parents, particularly in the case of prematurity. As mentioned above, the circumstances of preterm births, especially in more severe cases of prematurity, may increase the risk of developing postnatal psychopathology as well as parenting stress, which could negatively impact the quality of early dyadic exchanges. As the quality of the linguistic input serves important roles not only in facilitating the infant’s linguistic acquisition but also in supporting the development in several domains, including the socio-affective one, it is important to monitor the presence of typical or atypical IDS patterns during the first year, especially when they are associated with high levels of parenting stress or other risk factors. Considering the clinical implications of the findings, this study contributes to supporting the importance of implementing adequate psychological support for mothers during the early postpartum period, when needed, in the case of preterm infants with lower birth weight or high parenting stress. In fact, the possibility of perceiving a supportive and adequate environment, both at the hospital and after the discharge of the infant, may facilitate the co-construction of sensitive dyadic interactions, support parenting functions, and foster the development of mother–infant bonding, with improvement in infant development [76,77].

## Figures and Tables

**Figure 1 healthcare-12-00401-f001:**
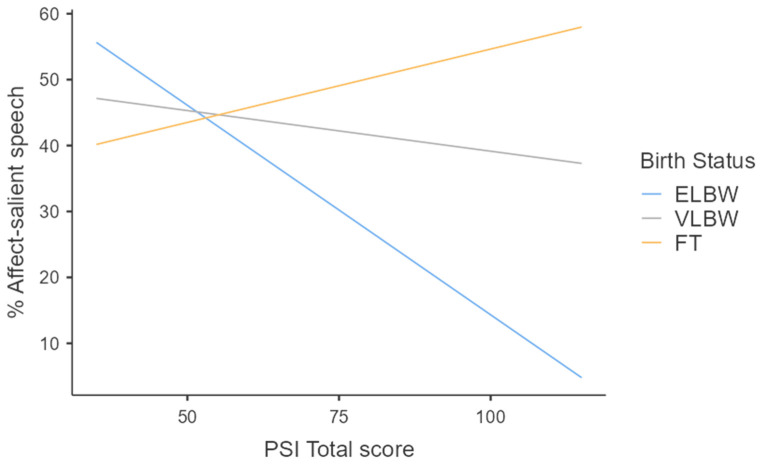
Effect of interaction between birth weight and maternal parenting stress on affect-salient speech at 3 months.

**Figure 2 healthcare-12-00401-f002:**
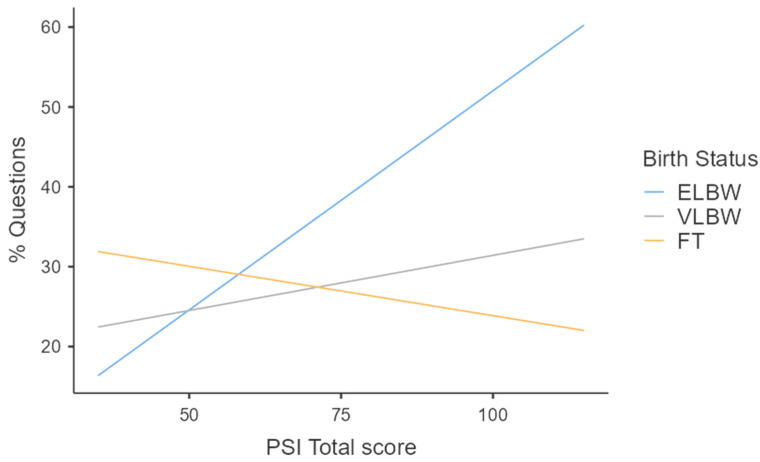
Effect of interaction between birth weight and maternal parenting stress on questions at 3 months.

**Figure 3 healthcare-12-00401-f003:**
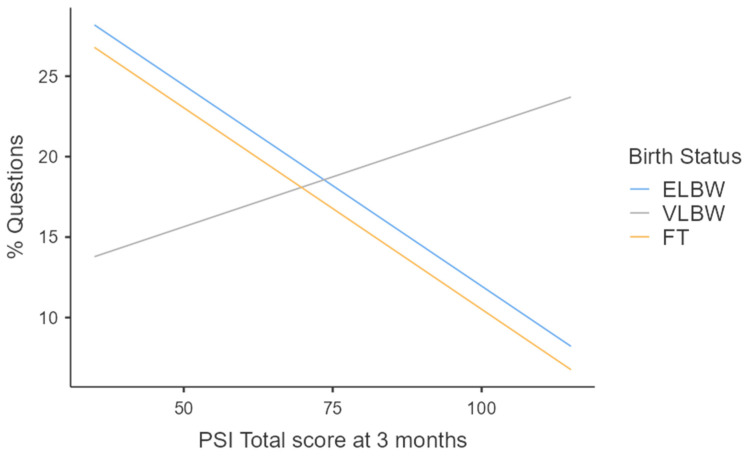
Effect of interaction between birth weight and maternal parenting stress at 3 months on questions at 9 months.

**Table 1 healthcare-12-00401-t001:** Participant characteristics.

		FT (N = 40)	VLBW (N = 30)	ELBW (N = 30)	F/X^2^	*p*
**Maternal variables**	Age	33.8 (4.77)	36.2 (5.17)	34.8 (4.66)	2.09	0.129
	Years of education	14.9 (2.97)	13.4 (2.50)	12.4 (3.49)	6.12	0.003 **
	Marital status ^a^				2.59	0.274
	Married/cohabit	34 (85)	29 (96)	27 (90)		
	Other	6 (15)	1 (4)	3 (10)		
	Parity ^a^				15.1	<0.001 ***
	Nulliparous	39 (98)	19 (63)	26 (86)		
	Multiparous	1 (2)	11 (37)	4 (14)		
	Working condition ^a^				2.13	0.345
	Employed	37 (93)	25 (83)	28 (93)		
	Unemployed	3 (7)	5 (17)	2 (7)		
**Infant variables**	Gestational age at birth (in weeks)	40.0 (1.79)	29.2 (2.55)	26.9 (1.71)	425	<0.001 ***
	Birth weight (in grams)	3445 (438)	1241 (148)	840 (150)	780	<0.001 ***
	Hospitalization days	2.15 (0.36)	35.7 (11.7)	63.6 (16.1)	279	<0.001 ***
	Infant’s mean age at T1 ^b^	3.00 (0.25)	3.05 (0.18)	3.09 (0.21)	1.65	0.197
	GMDS-R total score at T1 ^c^	114 (8.54)	108 (8.05)	107 (8.88)	7.80	<0.001 ***
	Infant’s mean age at T2 ^b^	9.5 (0.25)	8.9 (0.31)	9.0 (0.23)	61.3	<0.001 ***
	GMDS-R total score at T2 ^c^	103 (9.27)	99 (9.11)	100 (7.58)	2.63	0.077

Data are expressed as means (and standard deviations in parentheses) for interval data. ^a^ Number (and % in parentheses) for categorical data. ^b^ Corrected age for PT infants. ^c^ GMDS-R was administered and scored considering chronological age for FT infants and corrected age for PT ones. ** *p* < 0.01. *** *p* < 0.001.

**Table 2 healthcare-12-00401-t002:** Mean and standard deviation of maternal PSI scores at 3 and 9 months.

	T1			T2			Between Effect		Within Effect		Interaction	
	FT (N = 40)	VLBW (N = 30)	ELBW (N = 30)	FT (N = 40)	VLBW (N = 30)	ELBW (N = 30)	F (2, 97)	*p*	F (1, 97)	*p*	F (2, 97)	*p*
**PSI-SF Total Scores**	61.8 (15.5)	63.5 (19.1)	62.2 (14.6)	62.3 (14.2)	61.4 (24.2)	57.9 (11.5)	0.20	0.819	2.33	0.130	1.19	0.307

Data are expressed as means (and standard deviations in parentheses) for interval data.

**Table 3 healthcare-12-00401-t003:** Mean and standard deviation of lexical, syntactic, and functional features of IDS at 3 and 9 months.

	Birth Weight			Time of Assessment		T1			T2			Between Effect ^b^		Within Effect ^c^		Interaction	
	FT	VLBW	ELBW	T1	T2	FT (N = 40)	VLBW (N = 30)	ELBW (N = 30)	FT (N = 40)	VLBW (N = 30)	ELBW (N = 30)	F (2, 97)	*p*	F (1, 97)	*p*	F (2, 97)	*p*
**Lexical and syntactic characteristics of maternal IDS**																	
**Word tokens**	219 (9.47)	191 (10.9)	208 (10.9)	228 (7.44)	184 (6.23)	243 (57.1)	208 (91.8)	234 (73.4)	195 (52.5)	175 (60.2)	182 (73.6)	0.30	0.739	0.55	0.459	0.48	0.621
**Word types**	88.7 (3.21)	76.6 (3.71)	77.9 (3.71)	85.8 (2.45)	76.4 (2.20)	95 (21.9)	77.4 (24.1)	85 (27.2)	82.5 (21.5)	75.9 (20.8)	70.8 (23)	0.17	0.842	0.82	0.369	1.71	0.187
**MLU**	3.15 (0.08)	2.95 (0.09)	2.92 (0.09)	3.04 (0.06)	2.98 (0.05)	3.14 (0.66)	2.96 (0.59)	3.01 (0.57)	3.16 (0.51)	2.95 (0.57)	2.83 (0.64)	0.75	0.389	2.40	0.096	1.16	0.317
**Functional characteristics of maternal IDS**																	
**Affect-salient speech ^a^**	37.3 (1.76)	38.5 (2.03)	36.4 (2.03)	42.7 (1.46)	32.2 (1.29)	46.1 (13.3)	43.6 (13.9)	38.3 (16.4)	28.6 (11.9)	33.4 (12.9)	34.5 (13.7)	0.26	0.769	43.90	<0.001 ***	6.64	0.002 **
**Labeling ^a^**	2.29 (0.43)	2.94 (0.49)	2.54 (0.49)	1.98 (0.37)	3.20 (0.38)	0.81 (1.75)	2.64 (4.52)	2.48 (4.61)	3.77 (3.42)	3.24 (3.90)	2.59 (3.99)	0.50	0.607	5.60	0.020 *	3.19	0.045 *
**Descriptions ^a^**	19 (1.02)	18.7 (1.18)	16.4 (1.18)	15.5 (0.74)	20.6 (0.99)	16.7 (8.46)	15.8 (7.24)	14 (5.72)	21.3 (9.28)	21.7 (10.5)	18.8 (10)	1.17	0.314	3.21	0.076	0.14	0.873
**Questions ^a^**	25.9 (1.22)	22.2 (1.41)	26.9 (1.41)	28.8 (1.22)	21.2 (0.88)	26.4 (11.2)	28.6 (9.15)	31.3 (15.9)	23.2 (10.3)	17.9 (5.45)	22.6 (9.02)	2.88	0.061	12.40	<0.001 ***	2.00	0.141
**Directives ^a^**	6.94 (0.92)	7.69 (1.1)	8.49 (1.1)	5.38 (0.62)	10 (0.9)	3.46 (3.76)	5.8 (7.77)	6.89 (9.62)	10.4 (9.41)	9.59 (6.71)	10.1 (9.94)	0.61	0.547	21.92	<0.001 ***	1.51	0.226
**Attention-getter ^a^**	6.54 (0.79)	7.26 (0.92)	7.35 (0.92)	3.91 (0.52)	10.2 (0.77)	2.59 (4.41)	4.14 (4.46)	5 (6.44)	10.5 (7.36)	10.4 (6.96)	9.96 (8.58)	0.28	0.755	57.59	<0.001 ***	1.33	0.269

^a^ Data are expressed as means (and standard deviations in parentheses) for interval data. ^b^ The between-group effect refers to the differences among birth weight groups. ^c^ The within-group effect refers to the differences across the time of assessment. * *p* < 0.05. ** *p* < 0.01. *** *p* < 0.001.

**Table 4 healthcare-12-00401-t004:** Regression models for the prediction of IDS features at 3 months.

	R^2^adj	F (5, 94)	*p*	T	β	t	*p*
Affect-Salient Speech	0.15	4.21	0.002 **				
Birth weight 1 ^a^				−26.43	14.09	−1.88	0.064
Birth weight 2 ^b^				−42.31	14.07	−3.01	0.003 **
PSI_3 months				−0.64	0.17	−3.67	<0.001 ***
PSI_3 × Birth weight 1				0.51	0.22	2.35	0.021 *
PSI_3 × Birth weight 2				0.79	0.22	3.63	<0.001 ***
Questions	0.14	4.19	0.002 **				
Birth weight 1				20.45	11.76	1.74	0.085
Birth weight 2				41.02	11.74	3.49	<0.001 ***
PSI_3 months				0.55	0.14	3.80	<0.001 ***
PSI_3 × Birth weight 1				−0.41	0.18	−2.26	0.026 *
PSI_3 × Birth weight 2				−0.71	0.18	−3.87	<0.001 ***

Note: ^a^ Birth weight 1 = ELBW vs. VLBW; ^b^ Birth weight 2 = ELBW vs. FT. * *p* < 0.05. ** *p* < 0.01. *** *p* < 0.001.

**Table 5 healthcare-12-00401-t005:** Regression models for the prediction of IDS features at 9 months.

	R^2^adj	F(8, 91)	*p*	T	β	t	*p*
Affect-Salient Speech	0.12	2.63	0.012 *				
Birth weight 1				−1.02	14.56	−0.07	0.944
Birth weight 2				−18.81	15.21	−1.24	0.219
PSI_3 months				−0.53	0.18	−2.88	0.005 **
PSI_3 × Birth weight 1				0.26	0.26	1.02	0.310
PSI_3 × Birth weight 2				0.44	0.27	1.65	0.102
PSI_9 months				0.23	0.23	1.00	0.319
PSI_9 × Birth weight 1				−0.28	0.28	−1.02	0.311
PSI_9 × Birth weight 2				−0.25	0.31	−0.80	0.428
Questions	0.20	3.67	<0.001 ***				
Birth weight 1				−12.40	9.658	−1.294	0.202
Birth weight 2				−21.32	10.04	−2.122	0.037 *
PSI_3 months				−0.250	0.121	−2.063	0.042 *
PSI_3 × Birth weight 1				0.374	0.171	2.190	0.031 *
PSI_3 × Birth weight 2				−0.708	0.176	−0.004	0.997
PSI_9 months				0.202	0.153	1.319	0.190
PSI_9 × Birth weight 1				−0.248	0.180	−1.377	0.172
PSI_9 × Birth weight 2				0.329	0.207	1.590	0.115

Note: Birth weight 1 = ELBW vs. VLBW; Birth weight 2 = ELBW vs. FT. * *p* < 0.05. ** *p* < 0.01. *** *p* < 0.001.

## Data Availability

The authors agree to make the data presented in this study available on request from the corresponding author. The data are not publicly available due to privacy and ethical reasons.

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
