# Peer review of "Infant-Directed Speech to Preterm Infants during the First Year Postpartum: The Influence of Preterm Birth Weight and Maternal Parenting Stress"

_healthcare, 2024, doi:10.3390/healthcare12030401_

Round 1

Reviewer 1 Report

Comments and Suggestions for Authors

You are writing about early language, and yet you have not really defined language and its main components, i.e. communication development using gaze, gesture, targeted vocabulary use, simplified and repetitive language, intonation and being a responsive interactor (Harding et al, 2019). Harding et al ., (2019) write about the limited number of papers that consider language and communication beyond bonding and skin to skin, and that the linguistic aspects need to be stressed more. Please add this citation : Harding, C., Levin, A., Crossley, S. L., Murphy, R., & Van den Engel–Hoek, L. (2019). Effects of early communication intervention on speech and communication skills of preterm infants in the neonatal intensive care unit (NICU): a systematic review. Journal of Neonatal Nursing25(4), 177-188.

Please improve discussing LANGUAGE in your literature review and add these following points and citations:

Children born preterm have a higher risk of developing speech, language and communication problems: Johnson, S., Evans, T. A., Draper, E. S., Field, D. J., Manktelow, B. N., Marlow, N., ... & Boyle, E. M. (2015). Neurodevelopmental outcomes following late and moderate prematurity: a population-based cohort study. Archives of Disease in Childhood-Fetal and Neonatal Edition100(4), F301-F308.

The lower the birth gestational age, the higher the risk of speech, language and communication problems:Zambrana, I. M., Vollrath, M. E., Jacobsson, B., Sengpiel, V., & Ystrom, E. (2021). Preterm birth and risk for language delays before school entry: A sibling-control study. Development and psychopathology33(1), 47-52.

 Less gesture and pointing can occur with preterm communication development, and this is a significant risk factor for developing communication problems: 

Sansavini, A., Guarini, A., Savini, S., Broccoli, S., Justice, L., Alessandroni, R., & Faldella, G. (2011). Longitudinal trajectories of gestural and linguistic abilities in very preterm infants in the second year of life. Neuropsychologia49(13), 3677-3688.

- Late acquisition of first words is a problem: 

Imgrund, C. M., Krueger, B. I., & Getejanc, C. (2022). Conversational speech and language skills of children born preterm and full term. Communication Disorders Quarterly, 15257401221088505.

- Prems with receptive language problems may develop shorter MLUs later :

Jansson-Verkasalo, E., Valkama, M., Vainionpää, L., Pääkkö, E., Ilkko, E., & Lehtihalmes, M. (2004). Language development in very low birth weight preterm children: a follow-up study. Folia phoniatrica et logopaedica56(2), 108-119.

- Parent stress inhibits early infant-parent interaction: 

Milgrom, J., Newnham, C., Martin, P. R., Anderson, P. J., Doyle, L. W., Hunt, R. W., ... & Gemmill, A. W. (2013). Early communication in preterm infants following intervention in the NICU. Early human development89(9), 755-762.

- It is really improtant that you add that the neonatal environment is noisy and bewlidering for parents, and that this is likely to inhibit language and communication interaction early on: 

Cardoso, S. M. S., Kozlowski, L. D. C., Lacerda, A. B. M. D., Marques, J. M., & Ribas, A. (2015). Newborn physiological responses to noise in the neonatal unit. Brazilian Journal of Otorhinolaryngology81, 583-588.

RATIONALE for the Griffiths III - can you be clearer why you used this assessment tool rather than something more language focused such as the Pre-School Language Scales (PLS-4)? 

Can you state why you didn't use speech and language therapists to facilitate the infant-parent interactions? This would have enabled much richer data, I think.

Can you comment on the fact that 5 minutes of recording would be sufficient for you to make a judgement on the interaction style? I am not sure it would.

Can you state how much information the parents were given about the video that would be undertaken , and how much information were parents given about the language aspect of the data collection?

Can you give a clear rationale inked to language development about your data collection ages?

Comments on the Quality of English Language

The manuscript requires a careful re-read , with some editing required. I will give a few examples here as to what needs to be addressed:

- Please use either "infants" or "babies". It is confusing drifting between the two terms.

- Abstract, page 1, line 3: You have an odd sentence, "Despite literature research....". Do you mean to say "Despite an extensive literature research...."?

- Introduction, page 1, first sentence - "Prematurity condition....". It's probably better to write "Prematurity is defined as...."Also, in line 5, use "motor areas" instead of "motor ones".

- Page 3, line 12, second paragraph - "Wide previous literature" - what do you mean? This needs re-phrasing.

- Page 6, line 6, Please use "Data were analysed", not "Data analysis was...."

There are other similar examples, so a very careful re-reading will need to be undertaken.

Reviewer 2 Report

Comments and Suggestions for Authors

Summary and Recommendation: Though this article presents some interesting data that addresses some of the identified gaps in the current literature, its generalizability is limited both by the factors that the authors delineate in the Discussion and by a number of other factors as well.  Moreover, some of the semantics require revision and a number of design and analysis factors require clarification and/or modification.

1)      Introduction

a)      Though somewhat on the long side, this section provides a reasonable review of pertinent literature and in pertinent background concerning the primary outcome variable, namely infant-directed maternal speech patterns.

b)      The section also identifies a gap in the current literature, four aims the study is designed to achieve, and a study hypothesis

c)      Several semantic issues arise here and in the Abstract that need to be addressed:

i)       One issue that arises throughout the manuscript is how to describe one of the key predictor variables, namely “severe preterm birth.” This turns out to be defined in terms of gestational age <32 wks and birthweight <1500g.  Thus, it does not include all very preterm births, since only those <32 wks who are also very low birth weight (VLBW, i.e. <1500 g) are included; on the other hand, it may not include all very low birth weight infants, since some infants >=32 wks who are SGA may not be included.  Inasmuch as births are further subdivided according to birth weight (ELBW vs VLBW), and term infants are defined both by birth weight and gestational age, it may be preferable to categorize this variable simply as “birth weight.”  In fact, the Abstract refers to “prematurity severity based on birth weight.”

(1)   In any case, the term “severe prematurity” is not the term usually used to characterize either extremely preterm infants (<28 wks GA) or very preterm infants (28 to <32 wks GA).  Furthermore, the “severity” of prematurity is based on both gestational age and birth weight for age.

(2)   Several terms are used to describe this variable, including “birth status” and “birth condition,” neither of which seem a propos .  “Birth weight” though imprecise would be preferable, or perhaps simply “birth group.”

ii)     The second sentence of the Abstract (lines 10-12) needs to be reworded.  “Though published research has explored how...interaction, limited research …..”

iii)   “Maternal input” is used in the abstract on line 12, but it is not clear whether this refers specifically to infant-directed speech by the mother or something broader.

iv)   “Prematurity condition occurs” in line 27 should be reworded “Prematurity is defined as gestational age at birth <37 wks…” 

v)      “Wide previous literature” in line 309 presumably refers to a large number of articles or a broad range of articles, but his should be clarified.

vi)   This section as well as subsequent sections defines age at follow-up in terms of “post-partum” age, but for preterm infants corrected age has been used, at least for some variables.  This needs to be clarified and consistent.   

2)      Materials and Methods)

a)      This is a prospective observational cohort study comparing 3 groups of infants to determine whether and to what extent birth weight group and/or maternal stress influenced the primary outcome of maternal infant-directed speech patterns at 3 and 9 months of age.

b)       There is a discrepancy between the total stud population of 100 in line 150 and the total of 90 in line 13; the former is presumably the correct number

c)      With regards to representativeness of the groups, more information is needed to determine how many parents were approached and consented vs. refused.

d)      The exclusion criteria do not mention that premature infants who did not survive to discharge were presumably excluded, not there any information about how many of these children were thus excluded.

e)      There is no mention of attrition after discharge, but it would appear that all of those recruited were seen at both 3 and 9 months of age.  Nonetheless, this needs to be made explicit, since many studies have some attrition, even within the first year.

f)       Line 169 indicates “corrected age was considered for PT infants.”  It is not clear what this means or how it was calculated. 

i)       A footnote to Table 1 indicates that age at T1 and T2 was “corrected for PT infants,” but it unclear whether corrected age was used anywhere else. 

ii)     For example, was it used in administration or scoring of the Griffiths developmental assessment.

g)      Concerning developmental assessments in lines 171-174, it is not clear i) whether more than one psychologist was involved in the assessments (and if so what was done to assess/assure inter-rater reliability), ii) whether the psychologist(s) was blinded to birth weight group (and if so how).  iii) Nor was the training/experience of the examiner specified.

h)      In line 178-179 some further description of the CHAT method would be helpful, since many readers may not be familiar with it.

i)       No information beyond birth weight is provided regarding newborn characteristics. Many factors beyond birth weight and/or gestational age have been shown to influence infant outcomes, including prenatal steroids and a range of post-natal morbidities (e.g. IVH, BPD, sepsis, among others).  Presumably these same factors would also influence level of maternal stress during and after NICU stay.

j)       As noted in limitations, post-partum maternal mental health symptoms other than stress were not included in this study (e.g. anxiety, depression) though they have been shown in other studies to influence subsequent parent-infant interactions.

k)      Only a total score for the Griffiths is reported and this assessment is only used to compare birth weight groups over time.  In any case, more needs to be said concerning derivation of total score, its psychometric properties, and the mean score and S.D. 

i)       Also, “developmental quotient” is mentioned in line 245, but this is not defined. 

ii)     It is also not clear how corrected age was used in calculating scale or total scores.

iii)   Scores for some of the component scales would be interesting to look at, notably personal/social and language proficiency.

l)       The difference between Word Tokens (number of words) and Word types (number of distinct words) is not clear.  One suspects that the latter refers to number of unique words, whereas the former includes repetitions of those unique words.  If so, that should be specified. 

m)   Line 257 indicates that “we settled ELBW as the reference group.” Presumably this means they “chose” ELBW as the reference group for the regression models displayed in Table 4, but this should be clarified.

3)      Results

a)      Table 1 compares participant characteristics among birth weight groups.

i)       Mean (SD) birth weight and gestational age should be shown for each group.

ii)     Corrected age at Times 1 and 2 is shown, but not months since discharge, which may also be relevant.

b)      Regarding baseline maternal variables that significantly differed among birth weight groups, i.e. parity and years of education, it is not clear how their association was further assessed and determined not to be significant and thus not necessary to include in further analyses.  This is not specified in lines 274-277 nor in the statistical section of Methods. This is even less clear with regards to the two infant variables that differed among groups.

c)      It is surprising there were differences on GMDS-R total scores among birth weight groups at 3 months but not at 9 months.

d)      It seems paradoxical that there were no differences among birth weight groups or over time in maternal stress, since the LBW/PT groups would be expected to have higher levels of stress.  If there is no difference, then it is unclear how stress can be a predictor of IDS, as shown in Tables 3 and 4, either by itself or in interaction with birth weight group

e)      It would help to relate “between effect” and “within effect” in Table 3 to birth weight and time variables.

f)       Lines 296-298 appear to apply only to affective utterances.

g)      Several instances of “neither….nor” are preceded by “no” or “not,” in which case the conjunctions should be “either…or” e.g. lines 284 and 314

h)      In line 361, “as well as” should be “followed by.”

4)      Discussion

a)      Rather on the long side

b)      Several semantic issues:

i)       “coherent” in most cases should be “consistent” e.g. line 415 and 445 and 474

ii)     “resulted to be” should be “appeared to be” e.g. line 409 or “was not” e.g. line 426

iii)   “Moving on” in 495 should be “Regarding”

c)      The allusion to vulnerable child syndrome in lines 435-436 is an important one and worth further assessment in future studies.

d)      The discussion of the study by Spinelli in lines 444-445 appears to contradict lines 446-447.

e)      It is unclear what “different trend” is referred to in line 464.

f)       In lines 498-500, would suggest rewording as follows:  “Similar to what happened with FT mothers, …..directed to ELBW infants; ….”

5)      Conclusion

a)      The first paragraph is overly broad in that the only mental health variable examined in this study was maternal stress, and the effect was modified by interaction with birth weight group.

b)      The authors correctly identify a number of important limitations of their study in section 4.5, but miss some others noted above.

c)      The importance of implementing adequate psychological support to mothers during the early post-partum period may be suggested by the data, but the suggestion is speculative.

Comments on the Quality of English Language

See detail above

Round 2

Reviewer 1 Report

Comments and Suggestions for Authors

This manuscript is greatly improved by the amendments. I have no further comments to make. Thank-you.

Author Response

Dear reviewer, thank you sincerely for your positive feedback. We are pleased that you took the time to explore the refined version of our work, and we highly value the insightful comments and suggestions you provided. Your input has been instrumental in refining and improving the quality of our work.

Reviewer 2 Report

Comments and Suggestions for Authors

The authors have adequately addressed most of the issues/recommendations from my initial review. A few corrections are still needed.  Line 36 the word "jet" should be omitted. Line 43 the verb should be plural "have" since "a number" refers to multiple items.  Line 300 "Birth" should be "birth".  Though the authors indicated that had added a footnote to Table 3 to further clarify what "between effect" and "within effect" referred to, I did not see such a footnote.  Finally, though the authors indicated they had changed multiple instances of "resulted to be" to "appeared to be" or "was not", multiple instances of "resulted to be" remain: lines 349, 381, 401, 411, 487, 522. 

Comments on the Quality of English Language

See above comments to authors.

Author Response

Dear reviewer, thank you very much for the positive feedback. We are pleased that the work has been improved. We really appreciate your valuable suggestions. We have addressed the typos you pointed out (lines 36, 43, 300), reworded “resulted to be” in “appeared to be” (lines 349, 381, 401, 411, 487, 522), and added a footnote to Table 3 to provide a clearer explanation of the between and within effects.